# Development of Textile-Based Strain Sensors for Compression Measurements in Sportswear (Sports Bra)

**DOI:** 10.3390/s24237495

**Published:** 2024-11-24

**Authors:** Aqsa Imran, Shahood uz Zaman, Mozzan Razzaq, Ayesha Ahmad, Xuyuan Tao

**Affiliations:** 1Department of Textile Engineering, National Textile University, Faisalabad 37610, Pakistan; draqsa@ntu.edu.pk (A.I.); mozzanrazzaq37@gmail.com (M.R.); ayeshaahmad0005@gmail.com (A.A.); 2École Nationale Supérieure des Arts et Industries Textiles—ENSAIT, ULR 2461—GEMTEX—Génie et Matériaux Textiles, University of Lille, F-59000 Lille, France

**Keywords:** active wear, strain sensor, conductive threads, compression gauging sensors, smart wearable, sportswear

## Abstract

Women sports wearer’s comfort and health are greatly impacted by the breast movements and resultant sports bra compression to prevent excessive movement. However, as sports bras are only made in universal sizes, they do not offer the right kind of support that is required for a certain activity. To prevent this issue, textile-based strain sensors may be utilized to track compression throughout various activities to create activity-specific designed sports bras. Textile-based strain sensors are prepared in this study using various conductive yarns, including steel, Ag-coated polyamide, and polypropylene/steel-blended threads. Various embroidery designs, including straight, zigzag, and square-wave embroidery patterns, etc., were created on knitted fabric and characterized for strain sensing efficiencies. The experiments concluded that strain sensors prepared from polypropylene/steel thread using a 2-thread square-wave design were best performed in terms of linear conductivity, sensitivity of mechanical impact, and wide working range. This best-performed sample was also tested by integrating it into the sportswear for proposed compression measurements in different body movements.

## 1. Introduction

Sports bras are designed to minimize breast injury and limit excessive breast movement during intense exercises. Support and comfort are essential qualities of sports bras [1,2]. Their purpose is to provide stability, support, and compression to the core tissues and muscles of the body and to improve blood circulation in the body [3,4,5]. The pressure applied by sports bras plays an essential role in their performance. If the pressure is too low, the bras will not serve their purpose of supporting the breasts during movement. Whereas, if the pressure is too high, it will cause serious health problems like discomfort, sore muscles, itchiness, pain, breathing difficulties, and irritation [6,7,8]. Therefore, to monitor the compression in sports bras, various sensors should be used.

Electronic devices that generate legible electrical signals in response to some external stimuli from the environment are known as sensors. These stimuli can include any physical, electrical, chemical, biological, or other properties, such as changes in pressure, temperature, or humidity of the surroundings [9,10]. Strain sensors are electronic devices that respond to changes in deformation by changing the magnitude of the flowing voltage in the sensors. Due to the change in voltage, the output of the circuit changes and hence the changes in the deformation can be monitored by measuring the output change [11].

Wearable applications for real-time monitoring and treatment have gradually changed the look of healthcare and wellness because of recent technological developments in smart flexible textiles, processing, and construction. These applications used textile strain sensors, which combine flexibility, conductivity, breathability, wearability, durability, and high stretchability with strain-detecting capabilities [12,13,14]. These textile-based strain sensors can be used to make knitted/woven fabrics and then be stitched directly on the textile. They can also be integrated into the fabric/garment in the form of a patch by stitch or embroidery technology [15,16,17]. The basic working principle of textile-based strain sensors is that these sensors respond electrically to changes in their molecular structures, and hence, the deformation of the garment can be measured [4,18,19].

Li et al. developed wearable 1D yarn strain sensors in which elastic polyurethane (PU) core, conductive Ag-nanoparticles, graphene-micro sheets composite sheath, and silicone encapsulation layer (PDMS) were used for its formation by dip coating and then the fabric was constructed by the double weave. They used the proto samples on various joint positions of volunteers for their use as human sports tracker applications. Although the prepared yarn sensor had good mechanical properties, the yarn’s durability needed further improvement. However, the use of these developed sensors has not been investigated in sportswear applications [20].

Park et al. fabricated a wearable strain sensor to monitor human respiration. This sensor was developed by lock stitching of textile substrate made by twisting silver plated and nylon yarns on Lycra and polyester yarns. Thread-based textile strain sensors (straight pattern) were tested to be highly sensitive but only to 10% strain [21]. However, these sensors were not used for sports activities.

Estrada et al. presented an alternative approach for the development of resistive strain sensors by using standard conductive yarns. A blend of polyester 80% and elastodiene 20% as a multifilament support yarn and polyamide with a 99% pure silver coating as the conductive yarn was used on Singer Futura XL-550 embroidery machine for two zigzag embroidery patterns overlapped having different heights and stitch density. The resistance response of the sensor was linear up to 40% of elongation. The Wash fastness test gave a reduction in the sensitivity of the sensor [22]. However, these prototypes were targeted to be used in healthcare applications, and their use in sports activities was not investigated in the study.

Jose et al. fabricated an embroidered resistive textile strain sensor based on the principle of superposition by using a stainless steel conductive thread embroidered onto a polyester–rubber elastic knit structure. The developed sensor demonstrated an average gauge factor of 1.88 ± 0.51 over a 26% working range, low hysteresis (8.54 ± 2.66%), and good repeatability after being pre-stretched over a certain number of stretching cycles. However, these sensors were intended to be used in mechatronic devices for robotics therapies, and their use in sportswear or other activity movement was not discussed [23].

Similarly, some researchers worked on the development of sportswear for some sensing activities. Shathi et al. developed a conductive, flexible, and washable sports bra for various human health monitoring applications. They used PEDOT: PSS material and the pad-dry-cure method was adopted for the development of electrodes. They claimed that developed electrodes have sheet resistance in the range that can be used for biomedical and health monitoring devices [24].

The literature review indicates that previous studies have been done mostly on developing textile-based strain sensors from plated/coated conductive threads. However, no research is performed on the blended conductive thread. Coated/plated threads have limitations in terms of thread extension and available variety, because they damage the upper coating layers. This issue can be resolved by using blended conductive threads that have the possibility of fiber slippage during extension without complete damage. Similarly, the shelf life of coated materials is very small, as the upper coating layer can be peeled out with some mechanical forces. Blended conductive threads can overcome these issues of reliability.

It is also observed that these developed sensors are not investigated for their use in sportswear for various body movements. Some researchers developed electrodes for sportswear, but the use of conductive threads for this purpose is limited. The use of these developed electrodes, typically for compression measurement, is also not investigated.

Sportswear, especially sports bra plays a significant role in the wearer’s comfort, health, and performance. Unfortunately, sports bras are only being developed in universal sizes, and different sports bras according to their end application have not been made, so it causes various issues in terms of comfort, performance, and health of the wearer. Therefore, in this study, textile-based strain sensors have been developed, by using blended conductive threads, which are intended to be used in sportswear, especially in bras. As a result, the mapping of compression during different physical activities can be realized. With this compression data, we can produce more comfortable bras for different activities. This study aims to use the blended conductive threads for the preparation of strain sensors so that issues highlighted due to the use of coated/plated conductive threads may be minimized. Developed sensors were examined for various properties, and these strain sensors were also investigated for their reliability and washability to claim that these sensors may be used in sportswear for compression measurement during various activities. The following sections describe the materials used in this study, experimental procedure adopted, and details of testing to be performed to verify the sensing abilities.

## 2. Materials and Methods

### 2.1. Material Selection

After reviewing the previous research on textile-based strain sensors, different conductive threads were selected for the development of this sensor. These threads include blended threads (polyester/steel and polypropylene/steel-blended threads), pure metallic threads (steel thread), and coated threads (Ag-coated polyamide thread). The blended threads were purchased from Bekaert, Belgium. Ag-coated polyamide thread was purchased from Shieldex (Statex Produktions-und Vertriebs GmbH, Germany). Three different materials were selected for the development of the samples, and these materials were selected based on the change in the values of the resistance with the application of the stress on the sensor. These three kinds of conductive yarns have been widely used in smart textile applications. For the silver-coated polyamide yarns, they are usually used for conductive thread, textile antenna, etc. Similarly, stainless steel and blended threads are also widely used in various smart textile applications, including transmission lines 10–12.

The physical properties of the conductive threads are determined by performing specific test methods using specific testing protocols, as mentioned in Table 1. The conductive steel thread was a filament, and therefore, its TPI, twist type, and the number of plies could not be determined.

The microscopic images of all the selected conductive threads were taken with an Optika Digital microscope (Italy) using test standard ISO-17751-1 [28], as shown in Figure 1. This figure image shows the silver-coated polyamide thread with three plies, and the coating can be seen as a metallic luster in the figure; the twist in the thread can also be seen in the figure. Figure 1b shows the steel thread having no twist; as it is the filament thread made up of the steel, its metallic luster can also be seen. Figure 1c shows a polyester steel blend having a specific quantity of steel fibers in the polyester thread; the twist in the thread can also be seen, while d shows the steel-blended polypropylene thread having a small amount of steel fibers blended in the thread, and the twist level can also be seen.

The next section explains about the experimental procedure adopted for the development of strain sensors and properties of conductive threads used in this research.

### 2.2. Experimental Procedure

Initially, the selected threads were tested on the customized tensile tester for resistance change against the applied strain (Figure 2). The polyester/steel-blended thread was not used in these experiments, as it has very high resistance, which was not detected by the customized tester (it can only measure up to 500 KΩ). It is also assumed that a resistance of more than 500 KΩ is not suitable for their use in e-textile sensing applications. The samples were developed from the remaining threads. Among the remaining threads, the polypropylene steel blend showed the highest change in resistance compared to the silver-coated polyamide and steel conductive threads (Figure 2). For all three types of threads, three repeated testing for each type of thread was performed, and measurement errors (Standard Deviation) are also plotted in Figure 2. The S.D. for all samples is less than 1.4 percent of the average values that can be claimed as reliable results. As the strain is increased, fiber slippage may occur in polypropylene/steel-blended thread that will cause the increase in electrical resistance. Similarly, in Ag-coated thread, strain will cause damage/cracks on the coated surface that will ultimately increase the electrical resistance of the samples. However, in the case of steel thread, no change was observed, because this thread was 100% steel. This thread had a less strain percentage compared to the other two threads, but no change in resistance was observed in that strain range.

Based on these initial results of conductive threads, three threads were further investigated for strain sensor development. These threads further experimented with various patterns to finalize the best possible efficiency of the sensors.

A customized tensile tester was developed for the measurement of change in resistance with the change in length. The sensors give the value of the change in the resistance, and this change will be used for the generation of signals. These signals, in turn, will be converted into the graphical representation of the level of strain using specific electronic circuits.

The microcontroller of this tensile tester was an Arduino Nano. This card has built-in ADC that converts the analog values into digital automatically. Therefore, there is no need for the extra ADC. In this project, the analog pin A0 of the Arduino is used to measure the input tension and converts it into the digital value of resistance. Figure 3 illustrates the electronic schematic of the tester setup developed for these experiments.

Strain sensors developed in this research needed to be tested for the changing of the resistance with the continuous change in the dimension of the sensor. Although various tensile testers are available, they were not able to fulfill the testing requirements of this research because of the real-time synchronization requirement between the evolutions of strain and electrical resistance. The available tensile testers do not provide the required synchronized values. We were unable to measure the precise value of the change in resistance with the change in the dimension of the sensor by using a digital multimeter and a tensile tester, as these devices were not synchronized. Therefore, a customized tensile tester was developed for the testing of the samples, which gave the change in resistance per 0.1 mm change in the length of the sample.

It was calibrated with a digital multimeter (Keithley 7510, Keithley Instruments, Ohio, USA) by comparing the resistance values of both the tester and the digital multimeter. Three samples of Ag-coated polyamide threads were cut at a length of 5 cm. These samples were tested for linear electrical resistance with both the customized tester and Keithley 7510. The customized tensile tester provided the same value of resistance, with less than 0.5% variation, at a given dimension, as the digital multimeter Keithley 7510. Therefore, it was concluded that the device is precisely calibrated and can give accurate measurements and is quite reliable. The detection of resistance is not only a parameter, but still, it is the main parameter for reliability, as discussed in various literature. The device can give measurements of resistance up to 500 Kohm. This is enough resistance that can be shown by a conductive thread, and it can be discussed in future work if any thread will be used having a resistance of more than 500 Kohm.

This device has one fixed end and another movable end. The testing samples were fixed in between these two ends with the help of jaws. It was placed in the laboratory and connected to the power supply of 12 V. The tests were carried out at 21 °C and 65% relative humidity. The Arduino controller in the device was connected to the computer, and the alligator clip extensions were connected to the knobs of the voltage divider for the resistance measurement as the length change occurs.

The testing condition is only for reference, and it can be modified according to the requirement, but the results will be accurate as per the requirement, as the conditions of testing will not affect the results, as is proven by testing the samples in both horizontal, as well as the vertical, directions; moreover, the sample stretching within this range is giving accurate results, and hypothetically, the sensor will not be that stretched in the end use; rather, it will experience lesser strain on it. The dynamic, multidirectional, and variable intensity loading can easily be analyzed using the current testing conditions, and the sensor will give precise results.

These samples were first set to the laboratory conditions for 24 h. Each sample was placed under the jaws with a 50 mm distance between the jaws, and the jaws were screw-tightened to hold the fabric firmly so that it may not slip from these jaws. The jaws hold the fabric by grabbing it in between the groves of the jaw base, which is fitted with the jaw tops. The samples were raveled 12.5 mm from both sides, and the alligator clips were used to hold these threads, as shown in Figure 4. The fabric was stretched for 40 mm at a rate of 0.5 mm/s, and the resistance was measured after each 0.1 mm stretching.

Various patterns were investigated using these conductive threads and explained in detail in the next section.

### 2.3. Pattern Selection

Patterns used for the development of strain sensors were selected based on their ease of movement on the application of the strain. Different sensor designs were developed to check the impact of the pattern designs on the sensor efficiency, as shown in Figure 5.

A zigzag would allow movement while stretching without any damage to the threads 19. However, with the single-thread zigzag design, it was recognized that the current will stop flowing if the thread is slightly damaged at any point as the electrical connection/circuit would be lost/broken. This would happen, because the design did not provide any alternative flow paths to the electric current of the strain sensor. Therefore, the number of threads in the same design was increased to increase the current flow paths in the design, so that, even if the conductive thread of the sensor gets damaged at one point, the current will be able to flow through the other path; hence, the sensor will not stop working. Various variations were made to check the effect of the number of threads on the efficiency, reliability, and durability of the strain sensors.

Another design was developed using a square-wave structure with the same number of threads as shown in Figure 5. This design may become more beneficial than a straight embroidered pattern, as a straight pattern can be deformed on applying the stresses, but this pattern has small portions of straight patterns that provide a better path and less probability of breaking the threads on the application of the strain. This design may be less appropriate than the zigzag pattern, but the comparison will conclude which design is preferable for the development of strain sensors. A comparison of various designs is discussed in Table 2

The designs underwent analysis to assess the flow of current within these sensors. The optimal current flow occurred in designs where the yarn paths did not intersect, reducing the length of the current pathway. Thus, these designs met this criterion effectively. Implementing these designs in the final application proved favorable, as the dynamic loading of the sensors would not impact the yarn, and it could readily expand and contract in response to strain on the sensor. Employing a straight design or any other alternative might not endure the dynamic loading experienced by the sensor.

Potential damage may arise if a design lacks adequate flexibility for expansion or contraction, unlike the zigzag and square-wave designs. For instance, the straight embroidery design could sustain damage from sudden dynamic loading. The designs finalized in this manuscript, however, demonstrate resilience against real-world dynamic loading.

The following section describes the sample development using these finalized designs with various conductive threads described above.

### 2.4. Sample Development

The design of the experiment for this study is mentioned in Table 3. All the samples of textile-based strain sensors were made on single jersey knit fabric with a sample size of 4 inches × 4 inches. The size of the sensor design embroidered on the samples was 3 inches × 0.7 inches. The sensors were made by a computerized two-headed embroidery stitching machine (CT-1202). All samples were prepared in the repetition of three identical samples for each, and then, average results were used in each type of testing.

The stitches were made by using a two-headed embroidery machine (CT-1202, JINYU machine, China) with a SPI of 8 to develop a loose structure of lock stitch 301 so that the embroidery design of the sensor could give precise values.

Figure 6 shows the samples for these six pattern designs using polypropylene/steel-blended thread

### 2.5. Measured Characteristics

Linearity and Repeatability

Linearity is the direct relation between resistance changes to the applied strain. Ideally, the resistance should change in proportion to the change in the length of the sensor. It can be determined by the visual inspection of the graph that, if the resistance change increases positively with the increase in strain, then the sample has high linearity [29]. These samples were also investigated for the stability test. The ratio of change in resistance to the change in length was calculated for a total of 50 times. The samples were tested 10 times a day with a gap of 30 min of relaxing time, and this practice was repeated for 5 days.

Working range

The working range of the sensor represents the strain limit until the strain sensor shows a linear response of resistance change to the strain applied. The working range can be determined by noting the strain range until the line in the graph is straight [29].

It is another term that describes the use of the sensor in end applications. The deformation occurring in the sensor can be measured up to a certain limit that is allowed by its working range. It is described in percentages concerning the original size of the sensor embedded in the compression garments.

Gauge factor

The gauge factor represents the sensitivity of the strain sensor, and it can be calculated by the following formula:

The Gauge factor represents the sensitivity of the strain sensor, which means that even small variations in the body strain would be detected easily. Therefore, if the sensor is sensitive, it will have a large resistance change even for a small applied strain. The human body undergoes small changes if it works for a longer time, and the sensor should be sensitive enough to capture these changes in the body without problems.

The following section discuss the results of the above-explained characteristics on the developed samples.

## 3. Results and Discussion

The change in the electrical resistance concerning the change in length of up to 80% of the samples in a direction perpendicular to the jaws of the customized tensile tester (horizontal direction) was measured. The results showed a decreasing trend of electrical resistance against the elongation for all samples and all pattern designs. Figure 7a shows the test results for the Ag-coated polyamide thread in a straight design. The same samples were tested in a direction parallel to the jaws of the customized tensile tester (vertical direction). The values of the resistance change against applied strain were observed to remain constant, with little fluctuations, as shown in Figure 7b. The graphical representation of the data shows that the change in length across the direction of the sensor does not change the resistance of the sensor. This is because the thread is not stretched in this direction, as strain in this direction only stretches the fabric, but the thread remains in its place. As a result, the straight pattern designs did not give reliable results in the vertical direction.

The samples made with six pattern designs show that the resistance values change (almost linearly) on the application of strain in both directions (perpendicular and parallel to the jaws of the tester), as shown in Figure 8. The same trend was observed in all samples.

The difference between the zigzag and square-wave designs and the straight embroidery pattern shows that these designs can change in resistance, irrespective of the direction of strain applied. Therefore, all three threads were tested, and the graphical representations of the effects of the applied strains are as follows:

### 3.1. Ag-Coated Polyamide Thread-Based Strain Sensors

All samples of Ag-coated polyamide thread, with different designs (A1–A6), showed a decrease in resistance when the strain was applied to it in the horizontal direction. This was due to the structure of the Ag-coated polyamide thread. When the sample was stretched (when the strain was applied), the coated plies of thread came into contact with each other, and therefore, the conductivity of the samples increased, whereas the resistance decreased, which is illustrated in Figure 9.

From all the samples of the strain sensor developed from Ag-coated polyamide thread, the samples with a two-thread zigzag design (A2), one-thread square-wave design (A4), and two-thread square-wave design (A5) showed better performance concerning linearity. The R square values for these samples were 0.92, 0.98, and 0.95, respectively.

### 3.2. Steel Thread (B)

Most of the samples of strain sensors developed from steel thread did not show any specific trend in resistance upon applying strain (increasing their length). Therefore, the resistance of the samples remained approximately constant in all designs, even when the strain was applied to them, as shown in Figure 10, so the steel thread is rejected for use in the development of the textile-based strain sensors.

### 3.3. Polypropylene/Steel-Blended Thread (C)

All samples of polypropylene/steel-blended thread, with different designs, showed a decrease in resistance when the strain was applied to it, as indicated in Figure 11. This was due to the structure of the blended polypropylene/steel thread, which has short steel fibers in it, as discussed previously. When the samples were stretched, the steel fibers in the thread came into contact with each other, which increased the conductivity of the samples and decreased their resistance.

From all the samples of the strain sensor developed from polypropylene/steel-blended thread, the samples having a two-thread zigzag design (C2), three-thread zigzag design (C3), and two-thread square-wave design (C5) showed the best results concerning linearity. The R square values for these samples were recorded as 0.91, 0.91, and 0.90, respectively.

### 3.4. Comparison of the Prepared Strain Sensors

After the rejection of the steel thread, two threads were compared with six selected designs, and various functional properties of the samples were determined, as shown in Table 4. These functional properties are used to determine the sensitivity, range, ease of calibration, and accuracy of the developed textile-based strain sensors.

Out of the 12 designs, Ag-coated polyamide with designs A2 and A5 and polypropylene/steel-blended thread with designs C3 and C5 were concluded to be the best following their linearity, gauge factor, working range, and resistance change per mm, as shown in Figure 12.

The designs with two threads (A2, A5, and C5) and three threads (C3) had the best results concerning linearity, working range, resistance change/mm, and gauge factor, for both threads. This was because, as the number of threads in the designs increased, the current flow paths increased as well. Due to more flow paths, the connections between the conductive threads in the design increased, increasing the conductivity, linearity, and sensitivity of the sensor. However, due to increased conductivity, the resistance values decrease with the application of strain. Therefore, the two-thread designs gave the optimum level of resistance. However, in the three-thread designs, the resistance change was relatively small, so the two-thread design was concluded to be the better design. One possible reason is that the number of threads in the unit area is increased beyond the limit, which reduces its strain efficiency and thread stretchability, because the threads are locked at multiple points.

Although zigzag designs showed a more linear trend (as they stretched evenly when the strain was applied) as compared to square-wave designs, the latter had more working range and were more sensitive (as they had more gauge factor), as shown in Figure 12.

The best-performing samples were concluded based on the results of the testing, while the literature review makes us assume that the designs of the sensors that we have concluded give the best performance results, and hypothetically, the concluded design for the samples will give the same conclusion from the future experiments. Previously, various researchers worked on the working range and gauge factors, e.g., Bozali et al. [30] claimed to achieve a working range of 40% and gauge factor of 1.19. Similarly, Zahri et al. [31] was stated to achieve a gauge factor up to 0.99 and working range of 10–40%. Both these values validate our strain sensors’ performance based on the referenced literature.

### 3.5. Impact of Working Temperature

These developed strain sensors are intended to be used in sports applications where these sensors’ integrated bras would be used in various temperature conditions, including indoor and outdoor, and in various atmospheric conditions in different parts of the world. With the temperature change, all the parameters of the sensors are affected. Based on these reasons, the impact of the working temperature was also investigated in these experiments. These investigations were carried out on two best-performing threads, as explained earlier. The impact of temperature can be described in terms of temperature coefficient. Normally, the temperature coefficient of resistance for metals has positive values that show that the resistance of the metals increases with the increase in temperature, but in the case of conductive threads, the material that is dominating out of the two materials shows its properties over the other one. The working range of the sensors increases to some extent, because the increase in temperature enhances the flow of electrons in the metals (Table 5). The resistance change per mm for silver-coated thread was increased, and for metal-blended polypropylene, it was decreased. In the first case, silver was dominating, and hence, its properties were dominated. In other cases, the polypropylene properties were dominant, because they were in higher proportion (Table 5). The same was true in the case of the gauge factor, as it increases where the resistance increases with the increase in temperature, and it decreases where the resistance decreases (Table 5).

### 3.6. Repeatability Assessment Test

Among the four best best-performing samples, A2, A5, C5, and C3, samples A5 and C5 were further investigated for stability tests. These samples were tested for change in resistance against change in the length. The samples were extended for 40 mm, and the change in resistance was calculated. This practice was repeated 10 times with 30 min of relaxing time. This process was repeated over 24 h until a total of 50 repetitions. Figure 13 explains the standard deviation of these samples for each day. In both samples, standard deviations are under a good range even after 50 times repeatability. The overall standard deviation for completing 50 repeats is also 0.82 and 0.88 for A5 and C5, respectively. Hence, we can claim that these samples have not deteriorated after 50 repeats.

### 3.7. Wash Analysis

To verify the reliability of the samples, they were washed up to five cycles in the Launder-o-Meter. Each washing cycle lasts 30 min at 40 °C without using any detergent or softeners. Three samples of A5 and C5 were used for this purpose. These samples sustained their properties, and the ratio of the change in length with the initial length was approximately 25 times (Figure 14). As the initial values for these samples were in the range of 15 ohms, the final values even after five washing cycles were less than 500 ohms, which is in the working range of the strain sensors.

Following these tests, the developed samples were integrated in the sports bra of real-world testing on a group of women with running activities. These are explained in detail in the following section.

### 3.8. Integration in Sportswear

A real-world case study was conducted using a polypropylene–steel blend thread with a 2-T square-weave design (referred to as C5) applied to a sports bra (Figure 15). Commercially available sports bras in sizes 32 (B, C, and D); 34 (B, C, and D); 36 (B, C, and D); and 38 (B, C, and D) were chosen for the study. The most effective C5 strain sensor was integrated into the right bust of the sports bra using standard stitching techniques. The right side was selected to minimize any discomfort caused by heart vibrations.

A cotton knitted bra was selected for this study to avoid the production of static charge. Cotton is assumed to delay the production of electric static charge and decays the produced charge quickly. Moreover, activewears/sport wears are supposed to be tightly fitted on the body, and position of the sensors in the sports bra also reduce the friction movements during the activities.

Following institutional ethical approval, sixty female volunteers from a university staff and student population (mean + SD: age 30 ± 11 years, height 5 ft. ± 0.5 ft., body mass 50 ± 15 kg) were recruited for the study via posters and verbal and email communication. All participants were aged over 18 years (range: 19 to 45 years), had not been pregnant or breast-fed within the last year, or had any previous breast surgery; no restrictions were put on the upper age limit or ethnicity of the participants recruited. Participants were given a verbal explanation of the procedures and an opportunity to ask questions. When fully briefed, participants gave written informed consent. Participants then completed a questionnaire, where they reported their age and current bra size. Out of sixty, twenty-four participants with bra sizes of 32 C, 34 C, 36 C, and 38 C were selected for the wear trial.

Participants were asked to put on these sports bras and do running activity on a treadmill (model F-1200, FLEXOR, China) at three levels of speed: 10 km/h, 15 km/h, and 20 km/h for 10 min. This practice was repeated for B and D cup sizes after taking measurements for the C cup sizes. It was not possible to share the participant’s images, but for readers’ understanding, a simulation of the movements at CLO 3D is attached in Figure 16.

These samples were also tested for change in resistance by variation in compression during the running activity. When cup size B was used by the women with size C, pressure was increased on the sensor, and hence, resistance was decreased, as explained in Section 3.3. In this case, the ratio of change in resistance from the original value was less than 1 due to a decrease in resistance after compression. Similarly, in the case of cup size D, there was minimal pressure on the sensor, and the resistance was greater than the C cup size. In the case of cup size D, the R/R’ values were greater than cup size C due to less change in resistance because of reduced compression. In all cases, C cup sizes have a negative change in resistance (resistance reduced), because even if women have cup size C, there will be some sensor compression that causes the reduction in electrical resistance. The standard deviations of these samples are less than 0.2 for all the samples that are in acceptable range. Figure 17 explains the change in resistance for all twelve samples of various band and cup sizes for running speeds of 10 km/h, 15 km/h, and 20 km/h. Figure 17d explains the accelerating and decelerating variations of the resistance change. As participants started exercising, body movements raised that increased the compression level on the sensor until they reached the maximum level with time. Similarly, when decelerating started, body movement ultimately reduced the compression, but again, it will take time to achieve a body resting position.

These samples were washed for five washing cycles, as explained in Section 3.7, and then, the same practice was repeated to check the change in resistance by wearing the bras of various sizes (Figure 18). Here, again, the same trend was shown; however, the resistance was increased due to some wear and tear due to washing cycles. Similarly, the standard deviation in all samples also increased notably. As washing gave some surface damage and resistance was also increased, as described in Figure 14, it caused some deviation in the samples from normal readings, and some abrupt changes in the resistance values were observed.

Finally, these developed sensors were investigated for comfort assessment, as described in the next section.

### 3.9. Comfort Assessment

Each participant wore a sports bra for half an hour daily over a week. Subsequently, they wore both larger and smaller sizes for a week each, immediately following the base size, to evaluate the compression and comfort levels. After seven days of consistent wear, participants completed a questionnaire regarding the comfort of the bras (Table 6). Their responses were utilized to generate graphs for analyzing the comfort levels of the bras.

The responses of the women of different professions are graphically represented (Figure 19) for all four base sizes, and the average values of the responses of women of all professions are represented against each question of the questionnaire (Table 6). A threshold level is shown in the graph by an orange horizontal line; all the responses are shown on the x-axis, while the ratings are shown on the y-axis. A comfort level of 3.5 was selected in the range of average to good comfort for better analysis of the data. If it is below average, we can’t use these results as a better comfortable sensor.

According to the graphical representation of the responses:The comfort level of the base size of the bra for a certain level of pressure applied by the sensor is quite satisfactory, as it is above the threshold level, showing that the bras with the base sizes are comfortable to wear and are significantly useful for daily use.The aesthetic appeal of the bra having a sensor in it is significantly more comfortable, as analyzed by the graphical representation of the responses, as they are above the threshold level.The comfort level of the sensor at the cup area of the base size of the bra for a certain level of pressure applied by the sensor is quite satisfactory, as it is above the threshold level, showing that the bras with the base sizes are comfortable to wear and are significantly useful for daily use.In the case of a small size bra as compared to the base size, the compression level is higher while the comfort level hardly reaches the threshold level; in this case, the responses vary to a large extent, but still, the comfort level is quite significant.In the case of a large size of bra as compared to the base size, the compression level is less while the comfort level crosses the threshold level, showing that the comfort level is highly significant.In the case of a small base size, say 32 C, more responses were of the stance that the sensor was a bit irritating, but this irritation rating was not quite significant, because it was much lower, and the comfort level in this case reached the threshold level, while the other three base sizes were significantly comfortable, as per graphical representation of the responses.In the case of a large size of bra as compared to the base size, the compression level was less while the irritation level crossed the threshold level, showing that the comfort level was highly significant. The irritation was not caused by the large size of the bra as compared to the base size.Graphical representation of the responses of the differences in pressure of the sensor area as compared to the other areas are not significant, which means that the sensor imparts pressure on the body without compromising the pressure of the bra.The comfort level is quite significant in the case of the increase in the induced sensor area as per the graphical representation of the responses.

In the case of the gauge factor, the calculation is done using the values of resistance of the sensors and the value of the change of resistance with the application of the strain to stretch it to a specific length, and this length change with the original length is used to calculate the gauge factor of the sensor.

## 4. Conclusions

This research has been done for the development of textile-based strain sensors intended to be used in compression garments, especially sportswear, as the real-time monitoring of the strain produced in sportswear during different physical activities. Some of the properties of the developed strain sensors that play an important role in the end application of the sensors were measured, such as gauge, linearity, and working range.

The results analysis indicates that, although the samples developed from Ag-coated polyamide thread have large working ranges, the sensitivity and linearity of the samples are low. Therefore, these samples are not suitable for the development of textile-based strain sensors intended to be used in sportswear, especially sports bras, as the sensors will not be able to detect minor changes in the body/breast clearly during different physical activities. However, sample A5 was used for further investigation for reference with C5 design. Samples made from steel thread did not show any specific trend of changing resistance concerning the applied strain, because this thread is made up of steel, which is a good conductor of electricity, and the stiffness of the steel limits its stretchability. Moreover, due to its hardness, this thread is not compatible with the human body, so it is not a suitable choice for embedding in sportswear, which is the intended end application. Apart from that, the customized tensile tester did not detect any resistance change to the applied strain in the samples developed from blended polyester/steel thread, as they have high resistance (up to 100 MΩ), which is out of the machine capacity/range. Hence, it is not suitable to develop textile-based strain sensors from this thread, as it will be expensive due to the use of a highly sensitive controller for its measurement.

Therefore, the analysis of the results shows that the two-thread square-wave design, developed from blended polypropylene/steel thread, is the most suitable choice among the developed samples for developing the textile-based strain sensor, which is intended to be used in sportswear, especially in sports bras. This was because said sample is highly sensitive and can detect even minor changes in the body easily, as it will have a large resistance change even when the produced strain is small (because it has a high gauge factor of 1.52, and the resistance change/mm is approximately 3.84) and as the working range of the sensor is up to 40%, so the sensor developed will be able to detect large deformations in the body as well. This best-performing design, C5, along with A5, was used for further investigation on linearity and washability. These designs show linearity in resistance change to the applied strain and, therefore, is an appropriate choice for the sensor development, as the calibration will be easy and the uncertainty in output scaling will be minimized.

The finalized design was then also investigated for wash reliability, and it is claimed that the strain sensors were in the working range even after five washing cycles. These sensors were also integrated into a sports bra to evaluate its performance for sportswear. The results verified that, as the compression increased in the strain sensor, resistance change was detected, which indicates its potential use in various sports activities to calculate the compression.

This research is quite significant in the fields of medicine and sports; during the treatment of different diseases related to muscles, nerves, or skin, compression garments are used and the amount of applied pressure is a key point to measure so that the body may not be affected by the extreme or lower pressure. It needs a moderate pressure for the best support to healing, and this can only be possible when the pressure is measured in real time, and then, this pressure can be used for medical purposes. While in sports, sportsmen need to move their muscles in different directions during different physical activities, and the muscles, the nerves, and the veins undergo a dimensional change; if this change is not controlled by some external assistance, this may exceed a certain level and may cause injuries, so a moderate amount of the pressure can be measured by using these research techniques in compression garments in real time, which will help to improve the health of the sportsmen and will assist the sportsmen’s body not to injure and will resultantly save a sportsmen from soreness of muscles, itchiness, and pain of the body tissues.

## 5. Future Work

This research was focused on comparing the functionalities of various available conductive threads for their use as strain sensors for compression garments, especially in sports bras. These developed strain sensors are suitable to be used in sportswear. However, further investigations on their integration for various sports activities and their mapping are the next level that should be achieved to claim that these conductive threads may be used to measure the strain values during various sports activities.

## Figures and Tables

**Figure 1 sensors-24-07495-f001:**
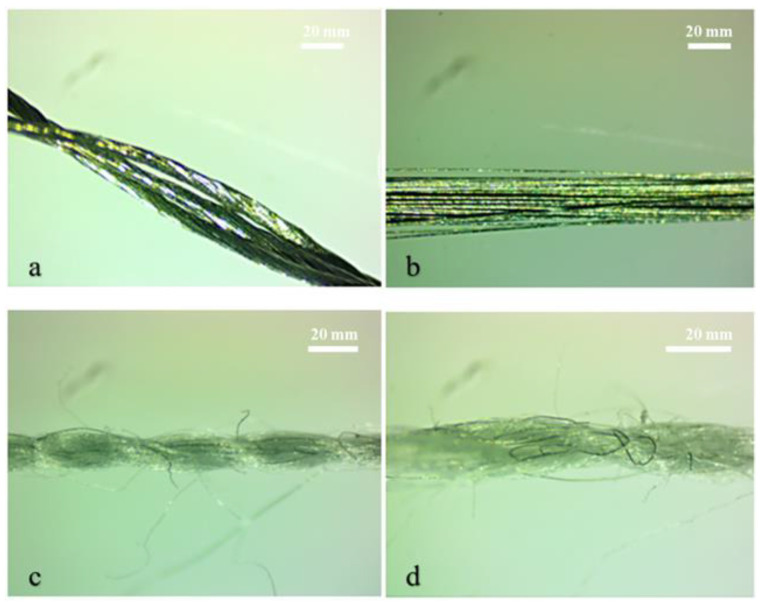
Microscopic images of conductive threads. (**a**) Ag-coated polyamide, (**b**) steel, (**c**) polyester/steel-blended, and (**d**) polypropylene/steel-blended.

**Figure 2 sensors-24-07495-f002:**
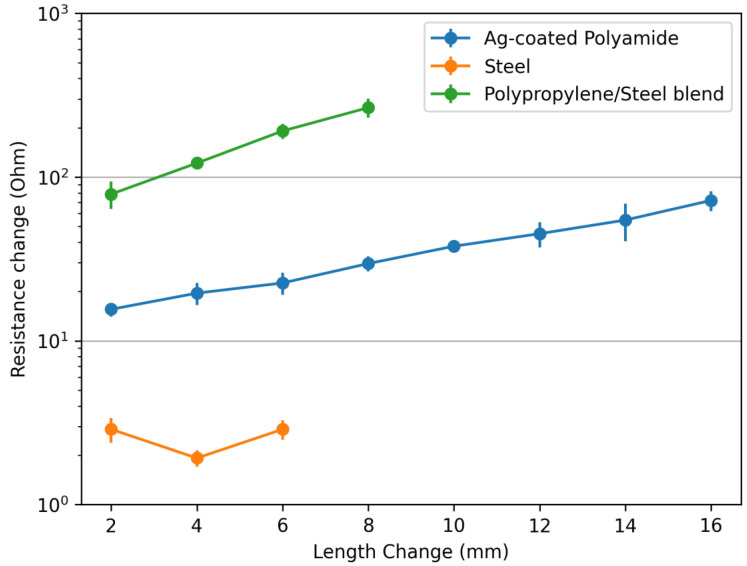
Evolution of elongation versus electric resistance.

**Figure 3 sensors-24-07495-f003:**
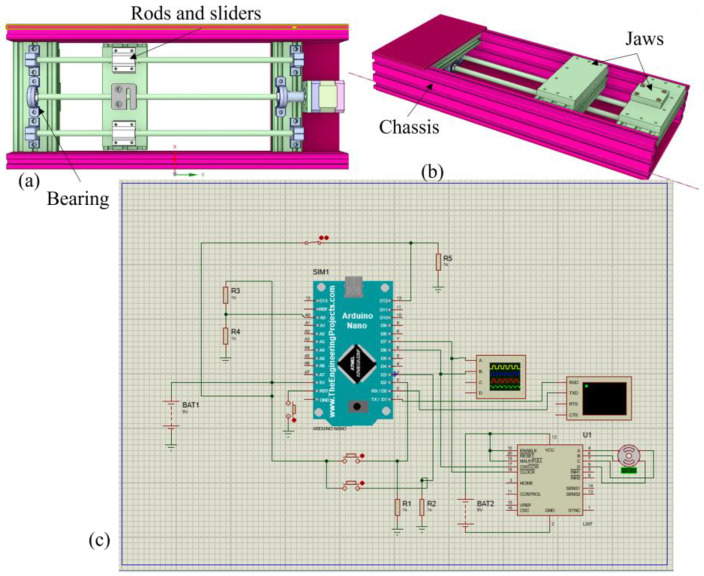
(**a**) Top view of a customized tensile tester. (**b**) Side view of a customized tensile tester. (**c**) Electrical schematic of the tester.

**Figure 4 sensors-24-07495-f004:**
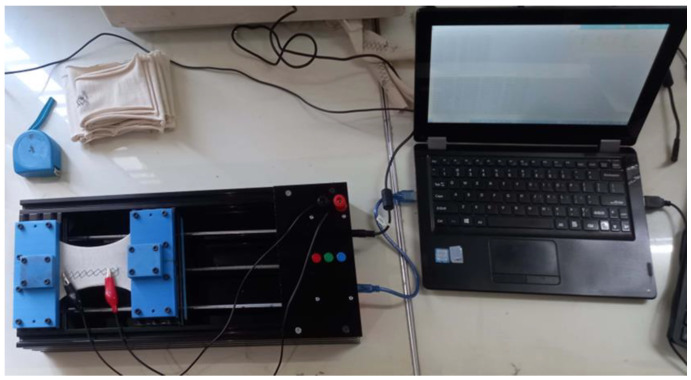
Experimental setup for testing.

**Figure 5 sensors-24-07495-f005:**
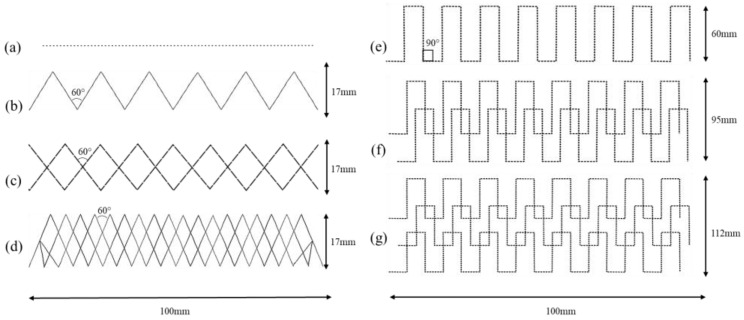
Pattern designs: (**a**) straight, (**b**) 1-thread zigzag, (**c**) 2-thread zigzag, (**d**) 3-thread zigzag, (**e**) 1-thread square-wave, (**f**) 2-thread square-wave, and (**g**) 3-thread square-wave.

**Figure 6 sensors-24-07495-f006:**
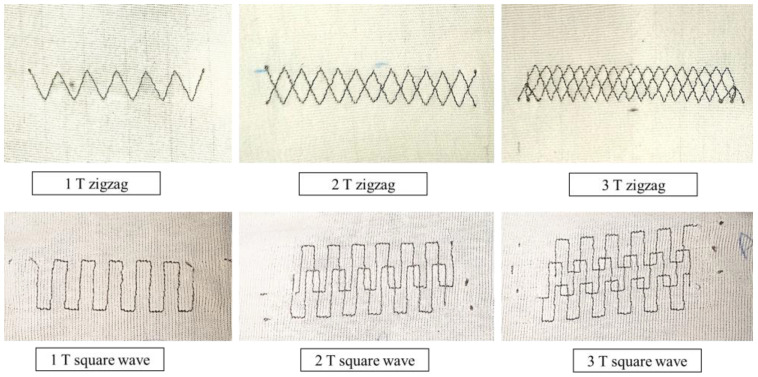
Proto samples of six pattern designs.

**Figure 7 sensors-24-07495-f007:**
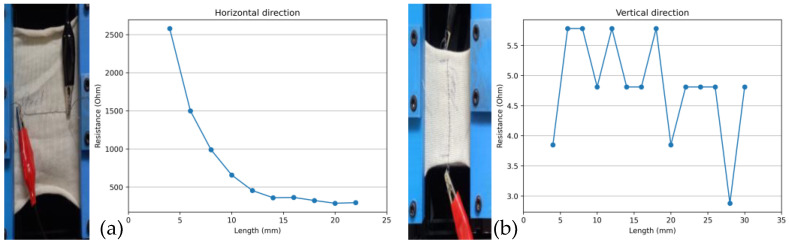
Sample testing of the straight design in (**a**) the horizontal and (**b**) vertical directions (sample code: A0).

**Figure 8 sensors-24-07495-f008:**
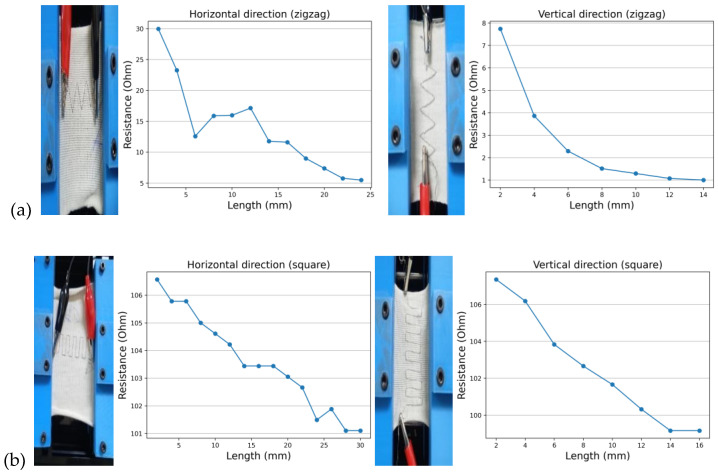
(**a**) Sample testing of the zigzag design in both directions (sample code: D1). (**b**) Sample testing of the square-wave design in both directions (sample code: A4).

**Figure 9 sensors-24-07495-f009:**
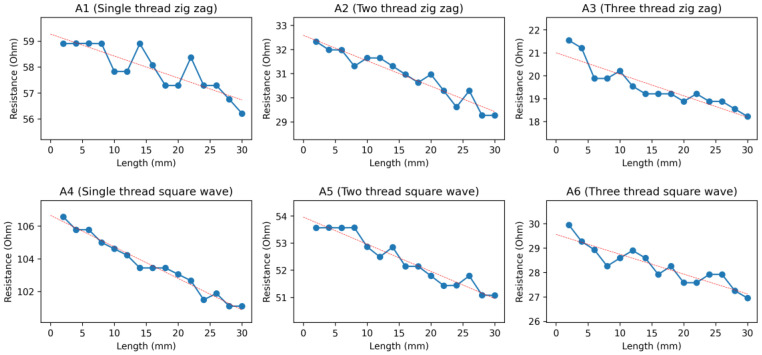
Graph of resistance against changes in length of the Ag-coated polyamide threads (Samples A1–A6). The red straight line is the fit linear line by linear regression.

**Figure 10 sensors-24-07495-f010:**
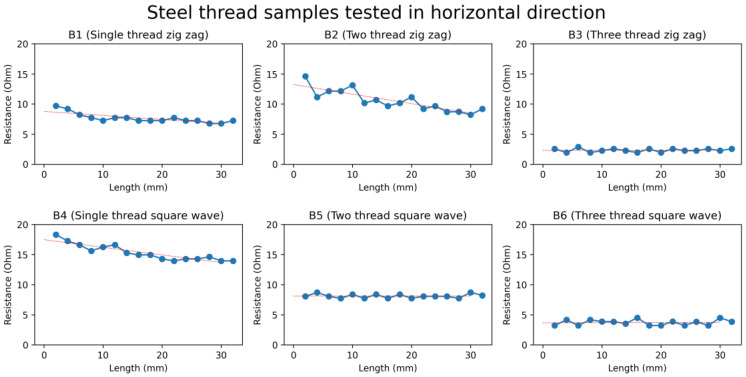
Graph of resistance against changes in length of steel threads. The red straight line is the fit linear line by linear regression.

**Figure 11 sensors-24-07495-f011:**
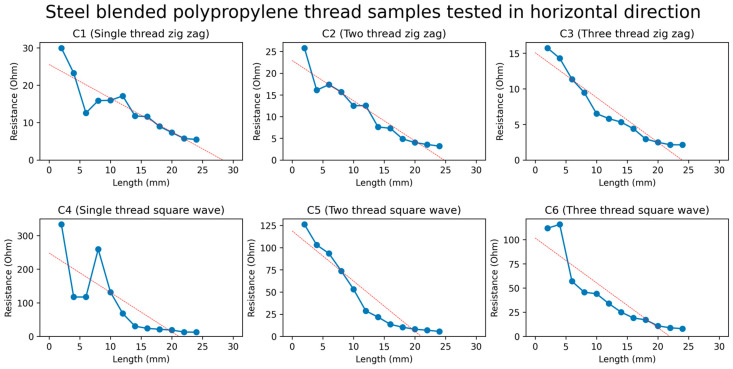
Graph of resistance against changes in length of polypropylene/steel−blended thread. The red straight line is the fit linear line by linear regression.

**Figure 12 sensors-24-07495-f012:**
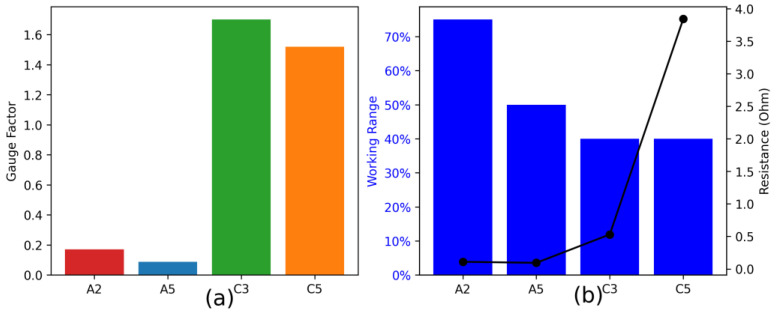
Functional properties of the selected samples. (**a**) Gauge factor of the selected strain sensors, and (**b**) working range and resistance change per mm of the selected sensors.

**Figure 13 sensors-24-07495-f013:**
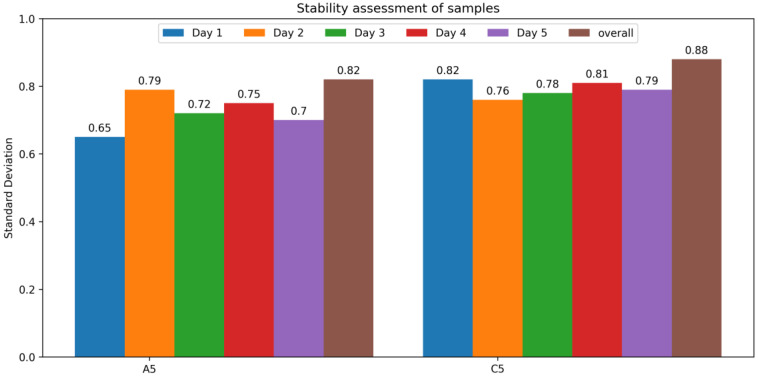
Repeatability assessment tests for samples A5 and C5.

**Figure 14 sensors-24-07495-f014:**
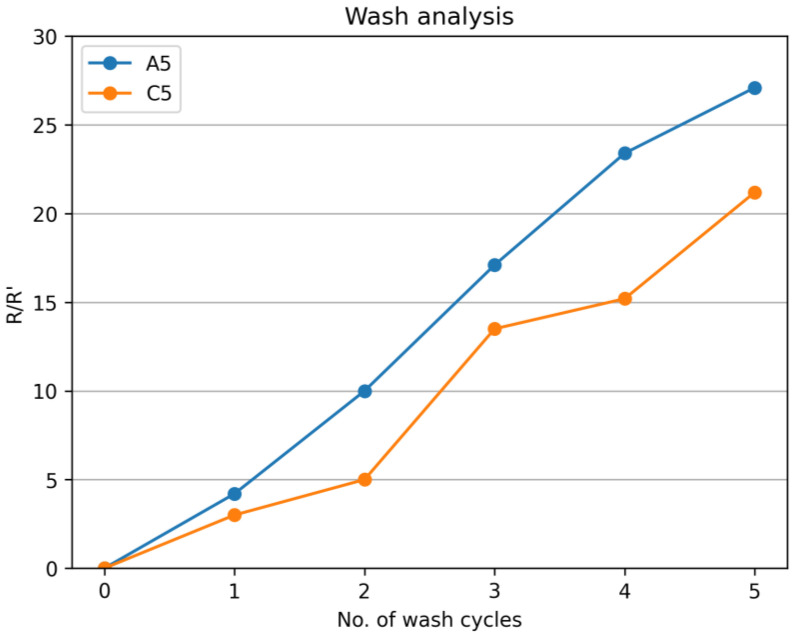
Wash analysis for samples A5 and C5.

**Figure 15 sensors-24-07495-f015:**
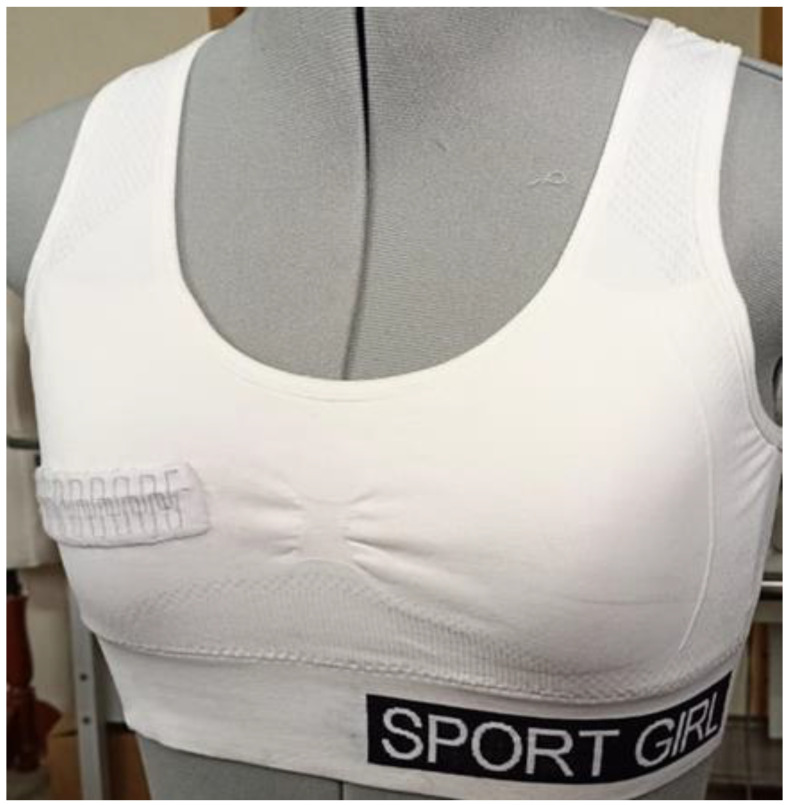
Prototype of the integrated sample used for the women wear analysis.

**Figure 16 sensors-24-07495-f016:**
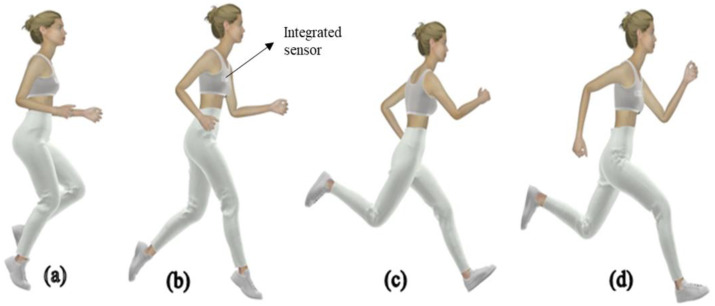
Simulation of the running activity in CLO 3D. (**a**–**d**) various running postures.

**Figure 17 sensors-24-07495-f017:**
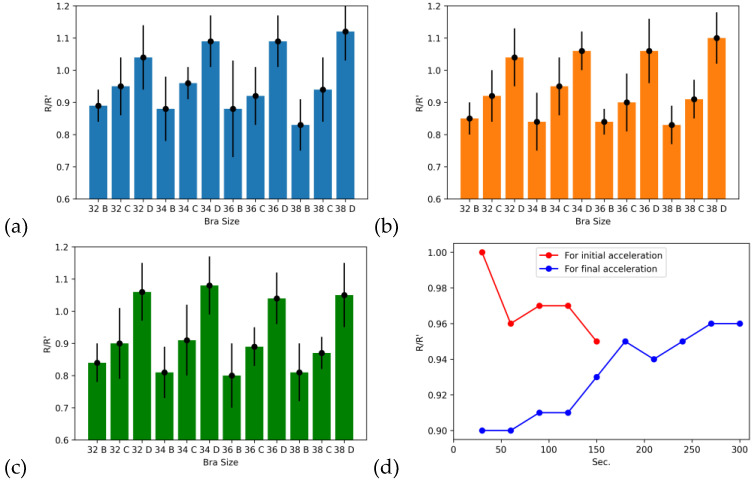
Resistance changes by changing the bra size of sportswear (**a**) at 10 km/h., (**b**) 15 km/h., and (**c**) 20 km/h. (**d**) Resistance change for accelerating and deaccelerating.

**Figure 18 sensors-24-07495-f018:**
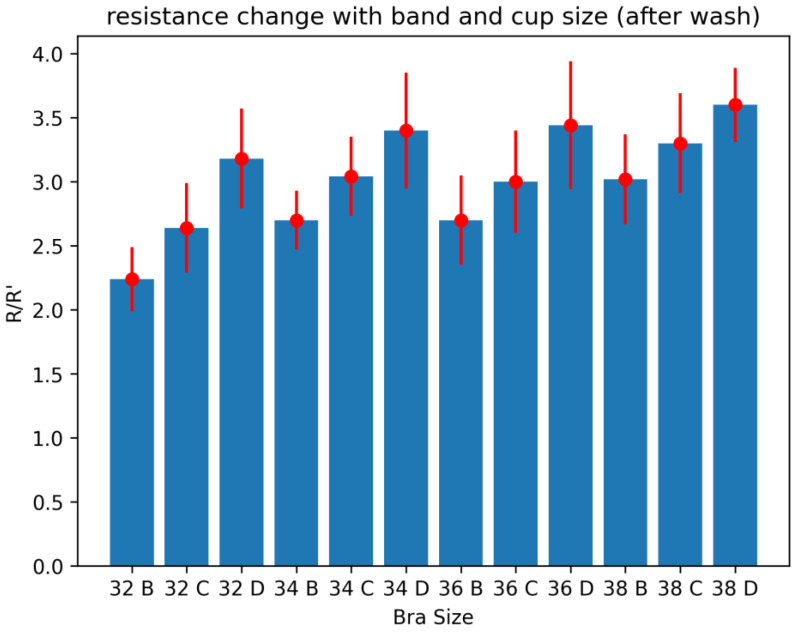
Resistance changes by changing the bra size of sportswear after 5 washing cycles.

**Figure 19 sensors-24-07495-f019:**
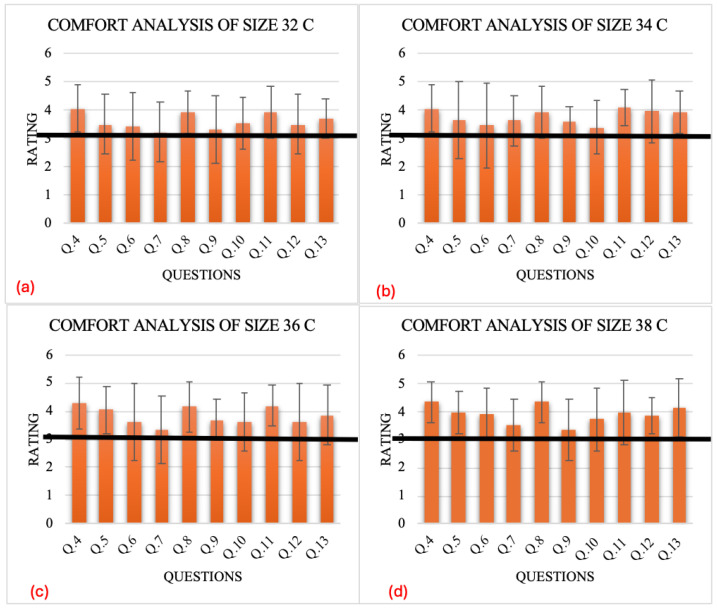
Graphical representation of the questionnaire for comfort analysis. (**a**) Size 32C (**b**) Size 34C (**c**) Size 36C (**d**) Size 38C.

**Table 1 sensors-24-07495-t001:** Physical testing and specifications of conductive threads.

Specifications	Device	Test Standards	Ag-Coated Polyamide Thread	Steel Thread	Polyester/Steel-blended Thread	Polypropyline/Steel-blended Thread
TPI (Twist/Inch)	Manual twist tester (direct method)	ASTM-D-1423 [25]	12.5	-	14.1	14.5
Twist type	Manual twist tester (direct method)	ASTM-D-1423	Z	-	S	S
Thread ply	Manual twist tester (direct method)	ASTM-D-1423	3	-	2	2
Count (Nec)	Weighing balance	ASTM-D-1059 [26]	19.69	4.88	15.36	14.92
Tensile strength (N)	Universal Tensile tester	ASTM-D-2256 [27]	7.78	19.09	11.03	6.83
Elongation (%)	Universal Tensile tester	ASTM-D-2256	34.60	1.52	11.73	4.54

**Table 2 sensors-24-07495-t002:** Comparison of three designs.

Straight	Zig Zag	Square-Wave
Can expand in only one direction.	Can expand in two directions	Can expand in two directions
Can not give an accurate result during the expansion in the diagonal direction.	Can give an accurate result during the expansion in the diagonal direction.	Can give an accurate result during the expansion in the diagonal direction.
Consume less yarn and ultimately show less change in resistance	Consume comparatively more yarn and show more change in resistance	Consume comparatively more yarn and show more change in resistance
The yarn’s embroidered structure has no stretchability, so can be broken on the application of high-strain	Embroidered structure has mobility of extending and contracting with fabric on the application and removal of the strain so the yarn design cannot be broken down.	Embroidered structure has mobility of extending and contracting with fabric on the application and removal of the strain so the yarn design cannot be broken down.

**Table 3 sensors-24-07495-t003:** Design of the experiment.

Run Order	Code	Conductive Thread	Design
1	A0	Ag-coated polyamide	Straight
2	A1	Ag-coated polyamide	1-T zigzag
3	A2	Ag-coated polyamide	2-T zigzag
4	A3	Ag-coated polyamide	3-T zigzag
5	A4	Ag-coated polyamide	1-T square-wave
6	A5	Ag-coated polyamide	2-T square-wave
7	A6	Ag-coated polyamide	3-T square-wave
8	B0	Steel	Straight
9	B1	Steel	1-T zigzag
10	B2	Steel	2-T zigzag
11	B3	Steel	3-T zigzag
12	B4	Steel	1-T square-wave
13	B5	Steel	2-T square-wave
14	B6	Steel	3-T square-wave
15	C0	Polypropylene/steel blend	Straight
16	C1	Polypropylene/steel blend	1-T zigzag
17	C2	Polypropylene/steel blend	2-T zigzag
18	C3	Polypropylene/steel blend	3-T zigzag
19	C4	Polypropylene/steel blend	1-T square-wave
20	C5	Polypropylene/steel blend	2-T square-wave
21	C6	Polypropylene/steel blend	3-T square-wave

**Table 4 sensors-24-07495-t004:** Determination of the functional properties of the samples from the graphs.

Sample ID	Working Range	Gauge Factor	Resistance Change (Ω) per mm
A1	10%	0.11	0.13
A2	75%	0.17	0.11
A3	20%	0.26	0.11
A4	80%	0.097	0.21
A5	50%	0.088	0.094
A6	10%	0.15	0.08
C1	20%	1.48	0.88
C2	30%	1.42	0.73
C3	40%	1.70	0.53
C4	10%	2.00	13.4
C5	40%	1.52	3.84
C6	30%	1.93	4.32

**Table 5 sensors-24-07495-t005:** Effect of temperatures on the working range, resistance change, and gauge factor.

Sample ID	Working Range			Resistance Change (mm)			Gauge Factor		
	21 °C	30 °C	37 °C	21 °C	30 °C	37 °C	21 °C	30 °C	37 °C
A2	75%	80%	85%	0.11	0.14	0.25	0.17	0.14	0.35
A5	50%	55%	60%	0.094	0.1	0.14	0.088	0.1	0.13
C3	40%	45%	60%	0.53	301	245	1.70	1.57	1.4
C5	40%	45%	50%	3.84	1163	941	1.52	1.34	1.3

**Table 6 sensors-24-07495-t006:** Questionnaire for real-time feedback for comfort assessment.

No	Question
1	Profession?
2	Bra size?
3	Age
4	Rate the comfort level of extracting compression level with the help of a sensor (1 for no comfort, 2 for little comfort, 3 for average comfort, 4 for good comfort, 5 for very high comfort)
5	Rate the aesthetic appeal of the strain sensor in the form of embroidery. (increasing from 1 to 5)
6	Rate the comfort level of the sensor in the cup area. (increasing from 1 to 5)
7	Rate the comfort level of the sensor of comparatively small size for more pressure. (increasing from 1 to 5)
8	Rate the comfort level of the sensor of comparatively large size for comparatively less pressure. (increasing from 1 to 5)
9	Rate the irritation caused by the sensor. (Decreasing from 1 to 5)
10	Rate the irritation caused by the sensor of comparatively small size for more pressure. (Decreasing from 1 to 5)
11	Rate the irritation caused by the sensor of comparatively large size for less pressure. (Decreasing from 1 to 5)
12	Rate the pressure difference at the area of the sensor in the bra as compared to the other areas. (Decreasing from 1 to 5)
13	Rate the comfort level of the bra cup with the increased sensor area. (increasing from 1 to 5)

## Data Availability

The original contributions presented in this study are included in the article. Further inquiries can be directed to the corresponding authors.

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
