# Peer review of "Development of Textile-Based Strain Sensors for Compression Measurements in Sportswear (Sports Bra)"

_sensors, 2024, doi:10.3390/s24237495_

Round 1

Reviewer 1 Report

Comments and Suggestions for Authors

This article discuss about sensors in sports bra. This topic is interesting. Results of this article could be commercialized. I have a few comments in below:

1) End of section 1. Please add inform about next sections for example: In section 2.1 has beed described ..., in section 2.2 has beed described ...

2) Add section Nomenclature as a last section. In this section should be added all symbols and constants with values.

3) In Section 2.2 should be add a figure, which ilustrate how measurement system has been constructed.

4) In Section 2.2 is information, that Arduino has been used. In Arduino is 10-bits Analog/Digital Converter. It has been used extra ADC converter? In the other hand, (if not) 10-bits AD conventer guarantee measurements with adequate accurancy? It should be describe.

5) Section 2.2: which multimeters have been used? How the results were compared?  It should be described more precisly. 

6) Section 2.2 Section 2.2: It is necessary calculate a measurements errors and uncertainties. Then is possible reliable compare results of measurements.

7) Figure 3 and Figure 5 need correction according with template. It shoukld be look as (an example)  graphs from Figure 2.

Comments on the Quality of English Language

I have not any comments.

Reviewer 2 Report

Comments and Suggestions for Authors

The authors developed a textile-based strain sensor for sports bras to monitor the strain of sportswear in different sports activities in real time. Impressively, the authors studied the development of textile-based strain sensors with hybrid conductive threads and demonstrated strain sensors prepared with Blended Polypropylene/Steel thread, a 2-thread square-wave design, and measured that the sensors performed best in terms of linear conductivity, mechanical shock sensitivity, and wide operating range, thereby minimizing the problems highlighted by the use of coated/plated conductive threads. Overall, this research is of great significance in the fields of medicine and sports. The manuscript can be accepted for publication after major revisions.
1. In the material selection section, it is recommended to explain whether the textile to which the sensor is attached and the clothes worn will generate static electricity? Will the generation of static electricity affect the signal acquisition of the sensor?
2. In the integration with sportswear section, it is recommended to measure the pressure changes of different frequencies of breathing when wearing a sports bra when not exercising?
3. In the integration with sportswear section, it is recommended to measure whether the pressure changes can be accurately measured when athletes wear the sports bra to exercise and accelerate or decelerate rapidly?
4. It is recommended that the authors include the location where the sensors are working in the picture of the running activity in Figure 15.

Comments on the Quality of English Language

The manuscript needs further polishing for better understanding by readers.

Reviewer 3 Report

Comments and Suggestions for Authors

This paper focuses on the development of textile-based strain sensors that can measure compression in sports bras, aiming to design activity-specific sports bras. The research team utilized various conductive yarns, including steel, Ag-coated polyamide, and polypropylene/steel blends, to create strain sensors in different embroidery patterns and compared their performances. The best-performing 2-thread square-wave design was integrated into a sports bra, enabling real-time monitoring of body changes during activities.

Some questions for consideration include:

  1. Additional tests might be needed to assess the long-term durability of the specific pattern design used for pressure measurement in practical settings like sports bras.
  2. How well the proposed sensors can withstand various exercise intensities or environmental conditions (e.g., sweat, temperature changes, relative humidity)?
  3. The performance of the strain sensor should be compared with those of previously reported papers, including the value of gauge factor and strain sensing range. 

Round 2

Reviewer 1 Report

Comments and Suggestions for Authors I am glad for Your answers. You did a great job. In my opinion this article is better now. But I have one more question: - you have written, that You used 10 bit ADC. Are You sure, that 10 bits ADC conventer is enough?

Author Response

Dear reviewer, many thanks for your review. The each analogy input of microcontroller Arduino Nano (https://store.arduino.cc/en-fr/products/arduino-nano) in this manuscript has built-in ADC conventer which provides 10 bits of solution (i.e. 1024 different values). Several previous research (cf. the following references) have proved that the use of these built-in ADC for sensing application is reliable and our experiment results in this manuscript have as well shown that the evolution of resistance change can be measured by using Arduino Nano with 10 bits analogy input. In this case, we consider that the buit-in 10 bits ADC conventer is enough. 

  1. P. Cheng, X. Zeng, P. Bruniaux, and X. Tao, “Design and research on multi-sensory comfort data acquiring of tight sportswear in motion,” Journal of Industrial Textiles, vol. 54, p. 15280837241258371, Jan. 2024, doi: 10.1177/15280837241258371.
  2. Wojciechowski, J.; Skrzetuska, E. Creation and Analysis of a Respiratory Sensor Using the Screen-Printing Method and the Arduino Platform. Sensors202323, 2315, doi:10.3390/s23042315.
  3. Nolden, R.; Zöll, K.; Schwarz-Pfeiffer, A. Smart Glove with an Arduino-Controlled Textile Bending Sensor, Textile Data Conductors and Feedback Using LED-FSDsTM and Embroidery Technology. Proceedings202168, 4, doi:10.3390/proceedings2021068004.

Reviewer 2 Report

Comments and Suggestions for Authors

The manuscript is suitable for acceptance for publication in its current form.

Author Response

Thank you for your previous suggestions to help us to improve this manuscript. We'd like to receive your decision of acceptance for this manuscript.

Reviewer 3 Report

Comments and Suggestions for Authors

The manuscript has been well-revised.

Author Response

(The authors gave the same response as above.)
